# Coaxial Drainage versus Standard Chest Tube after Pulmonary Lobectomy: A Randomized Controlled Study

**Massimiliano Bassi** [1,*] , **Emilia Mottola** [1] , **Sara Mantovani** [1] , **Davide Amore** [1] , **Andreina Pagini** [1] , **Daniele Diso** [1] , **Jacopo Vannucci** [1] , **Camilla Poggi** [1] , **Tiziano De Giacomo** [1] , **Erino Angelo Rendina** [2] , **Federico Venuta** [1] and **Marco Anile** [1]

[1] Division of Thoracic Surgery, Department of General Surgery and Organ Transplant "PARIDE STEFANINI", Policlinico Umberto I, Sapienza University of Rome, 00161 Rome, Italy; emilia.mottola@uniroma1.it (E.M.); sara.mantovani@uniroma1.it (S.M.); davide.amore@uniroma1.it (D.A.); andreina.pagini@uniroma1.it (A.P.); daniele.diso@uniroma1.it (D.D.); jacopo.vannucci@uniroma1.it (J.V.); camilla.poggi@uniroma1.it (C.P.); tiziano.degiacomo@uniroma1.it (T.D.G.); federico.venuta@uniroma1.it (F.V.); marco.anile@uniroma1.it (M.A.)

[2] Thoracic Surgery Unit, Sant'Andrea Hospital, Università La Sapienza, 00189 Rome, Italy; erinoangelo.rendina@uniroma1.it

\* Correspondence: massimiliano.bassi@uniroma1.it; Tel./Fax: +39-06-49970220

**Abstract:** Chest tubes are routinely inserted after thoracic surgery procedures in different sizes and numbers. The aim of this study is to assess the efficacy of Smart Drain Coaxial drainage compared with two standard chest tubes in patients undergoing thoracotomy for pulmonary lobectomy. Ninety-eight patients (57 males and 41 females, mean age $68.3 \pm 7.4$ years) with lung cancer undergoing open pulmonary lobectomy were randomized in two groups: 50 received one upper 28-Fr and one lower 32-Fr standard chest tube (ST group) and 48 received one 28-Fr Smart Drain Coaxial tube (SDC group). Hospitalization, quantity of fluid output, air leaks, radiograph findings, pain control and costs were assessed. SDC group showed shorter hospitalization (7.3 vs. 6.1 days, $p = 0.02$), lower pain in postoperative day-1 ($p = 0.02$) and a lower use of analgesic drugs ($p = 0.04$). Pleural effusion drainage was lower in SDC group in the first postoperative day (median $400.0 \pm 200.0$ mL vs. $450.0 \pm 193.8$ mL, $p = 0.04$) and as a mean of first three PODs (median $325.0 \pm 137.5$ mL vs. $362.5 \pm 96.7$ mL, $p = 0.01$). No difference in terms of fluid retention, residual pleural space, subcutaneous emphysema and complications after chest tubes removal was found. In conclusion, Smart Drain Coaxial chest tube seems a feasible option after thoracotomy for pulmonary lobectomy. The SDC group showed a shorter hospitalization and decreased analgesic drugs use and, thus, a reduction of costs.

**Keywords:** chest tubes; lobectomy; coaxial tube; health costs

## 1. Introduction

Chest tubes are routinely inserted after major thoracic surgery procedures and vary largely in literature in terms of size, numbers and timing of removal. Most of the surgeons prefer to leave two chest drainage tubes after thoracotomy for major pulmonary resections. However, it has been already reported that a single pleural drain after pulmonary lobectomy is safe and effective [1–4] but none of these studies is definitive.

The Smart Drain Coaxial (SDC) chest tube (Redax. Modena, Italy), is built with an internal lumen with distal bores for air evacuation and four external fluted channels for fluid drainage (Figure 1); this conformation allows to drain air and effusions along the entire length of the tube. This device has been already evaluated in a recent retrospective study [5], with promising results.

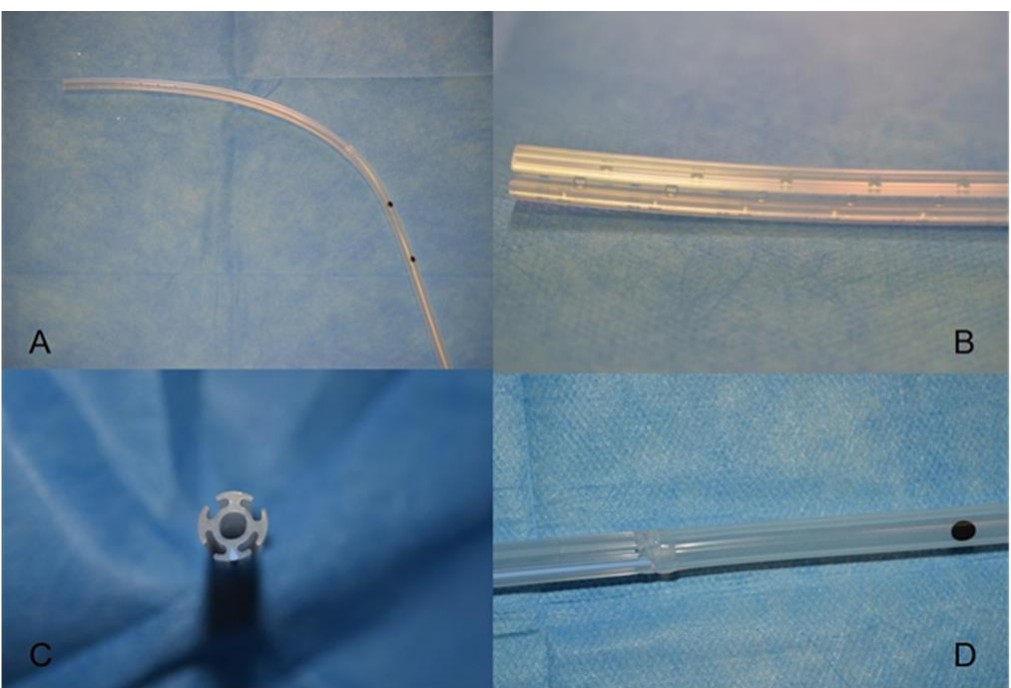

**Figure 1.** Smart Drain Coaxial tube (**A**). These tubes consist of four external fluted channels for fluids drainage and an internal section which allows separate air evacuation from appropriate distal bores (**B**–**D**).

In this regard, the aim of this study is to compare the use of two standard tubes (STs) with one SDC after thoracotomy for pulmonary lobectomy. The primary endpoint aims at verifying the efficacy of SDC in terms of fluid and air drainage evaluating the daily quantity of fluids collected and the presence of postoperative pneumothorax or pleural effusion at radiological imaging; the secondary endpoint was to assess incidence of complications after chest tubes removal, postoperative pain, hospital stay and costs compared to STs. We present the following article in accordance with the CONSORT reporting checklist.

## 2. Materials and Methods

### 2.1. Study Population

The study was approved by the institutional Ethics Committee of Policlinico Umberto I, Department PARIDE STEFANINI, informed consent was taken from all the patients and a trial registered in ClinicalTrials.gov (ID NCT04877925). Between February 2017 and April 2018, ninety-eight consecutive patients (57 M and 41 F, mean age 68.3 ± 7.4 years, median 69.0 ± 10.8) undergoing pulmonary lobectomy plus standard lymphectomy for lung cancer through lateral thoracotomy were included in the trial. Sample size has been calculated according to international literature on this topic and considering a confidence level at 95% and margin of error 5%. Inclusion criteria were: age more than 18 years; patient with lung cancer scheduled for open pulmonary lobectomy after a multidisciplinary team discussion. All patients were adequately informed and they have signed a consensus to participate. Exclusion criteria were: middle lobectomy, extended resections, minimally invasive lobectomies, previous ipsilateral thoracic surgery, induction chemo- and/or radiotherapy and patients who did not give consent to participate.

Patients were randomized into two groups using the block randomization procedure with a block size of 4: 50 patients received one upper 28 Fr and one lower 32 Fr standard chest tube (ST group) and 48 patients received one 28 Fr SDC tube (SDC group) both connected to a standard water seal. The result of randomization was communicated to the surgeon at the end of the surgical procedure, just before tube placement.

### 2.2. Clinical Data

Clinical and surgical variables were prospectively collected including: age, gender, comorbidity, smoking history, type of surgery, intensive care unit (ICU) length of stay (days), stage, chest drainage daily fluid output (mL), daily air leakage (+/−), postoperative pain score assessed by a visual analogue scale (VAS), administration of analgesic drugs, steroids and diuretics, therapy at discharge, radiologic evaluation of pleural effusion, pneumothorax or subcutaneous emphysema, postoperative hospital stay and incidence of complications. Regarding the fluid output, the daily effusion and the total collection entered the analysis with the aim to compare any difference between groups in day-by-day output and in larger periods of time (first 3 days and overall hospital stay).

Postoperatively, chest X-ray (CXR) was routinely performed on the first and third postoperative day (POD). A −20 cm $H_2O$ suction was applied at the end of the operation and maintained until the removal of chest tubes. Chest tubes were removed when pleural effusion was less than 300 mL/24 h in absence of air leaks for at least 24 h. In case of double tube drainage, they are always removed at the same time. CXR was also performed 24 h after removal of the drainage.

The quantity of pleural effusion was evaluated with CXR using a 4-grade scale, as reported by Mammarappallil and colleagues [6], at the first and third POD, and after tube removal: 0—normal costophrenic angles; 1—fluid meniscus below the hemidiaphragms; 2—fluid meniscus at the level of hemidiaphragms; 3—fluid opacity obscures the hemidiaphragms.

### 2.3. Statistical Analysis

Statistical analysis was performed according to international guidelines [7] using SPSS v.17 (SPSS Inc., Chicago, IL, USA). Categorical data are presented as number (*n*) and percentages (%); continuous data as median ± inter-quartile range (IQR) and mean ± standard deviation (SD). Fisher exact test was used to compare categorical data. Student's t test and Mann Whitney U test were used to compare normally distributed and non-normally distributed continuous data after Shapiro–Wilk test respectively. Sample size was calculated using Power up Microsoft software and considering a probability of type I error (alfa) 0.05 and power (1—beta) 0.95. A *p*-value ≤ 0.05 was considered statistically significant.

## 3. Results

Baseline characteristics and operative data are reported in Table 1. No statistical differences were found between the two groups in terms of age, sex, smoking history, comorbidity, side of operation and staging. All patients but two had a peridural catheter for postoperative pain management with infusion of ropivacaine 240 mcg and morphine 3 mg for 24 h at 2 mL/h. The remaining two patients received 2 mL/h intravenous infusion of Sufentanyl 4 mcg pro kg for the first 24 h. In addition, as standard management, paracetamol 1 g was intravenously infused three times per day (BASE group). Supplementary administration of analgesic drugs (i.e., ketorolac or morphine's bolus) was recorded (PLUS group).

Right upper lobectomy was performed in 34 (34.7%) patients, right lower lobectomy in 25 (25.2%), left upper lobectomy in 20 (20.4%) and left lower lobectomy in 19 (19.4%). All patients received systematic mediastinal lymph node dissection. Seventy-one patients had pulmonary adenocarcinoma (72.4%), eleven squamous cell carcinomas (11.2%), four atypical carcinoids (4.1%), two small cell lung cancers (2%), five (5.1%) other primary lung cancers and five pulmonary metastases from previous malignances (5.1%). Staging for primary lung cancer, according to the 8th edition of the International Association for the Study of Lung Cancer, was distributed as follows: stage IA 35 patients, IB 23 patients, IIA 6 patients, IIB 15 patients, IIIA 10 patients, IIIB 2 patients (N2 single station) and IVa 2 patients (treated single brain metastasis). In all the patients, the regular lobectomy plus standard lymphectomy was performed without intraoperative complications and showed no significant anatomical variances. Hemostatic agents such as fibrin sealants, hemostatic matrix or oxidized regenerate cellulose were applied based on intraoperative findings and surgeons' preference.

**Table 1.** Baseline characteristics and operative data. Data are *n*(%) or mean ± standard deviation.

|  | Overall | Standard | Coaxial | *p*-Value |
|---|---|---|---|---|
| *n* | 98 | 50 | 48 | |
| Age | 68.3 ± 7.4 | 69.0 ± 7.1 | 67.6 ± 7.8 | 0.21 |
| Male | 57 (58.2%) | 28 (56%) | 29 (60%) | 0.69 |
| Active Smokers | 73 (74.5%) | 38 (76.0%) | 35 (72.9%) | 0.66 |
| Lobe | | | | 0.54 |
| Left lower lobe | 19 (19.4%) | 10 (10.2%) | 9 (9.2%) | |
| Left upper lobe | 20 (20.4%) | 8 (8.2%) | 12 (12.2%) | |
| Right lower lobe | 25 (25.5%) | 15 (15.3%) | 10 (10.2%) | |
| Right upper lobe | 34 (34.7%) | 17 (17.3%) | 17 (17.3%) | |
| Histology | | | | 0.09 |
| Adenocarcinoma | 71 (72.5%) | 34 (34.7%) | 37 (37.8%) | |
| Squamous cell cancer | 11 (11.2%) | 9 (9.2%) | 2 (2.0%) | |
| Others | 16 (16.3%) | 7 (7.1%) | 9 (9.2%) | |
| Stage | | | | |
| IA | 35 (35.7%) | 13 (13.3%) | 22 (22.4%) | 0.29 |
| IB | 23 (23.5%) | 13 (13.3%) | 10 (10.2%) | |
| IIA | 6 (6.1%) | 3 (3.1%) | 3 (3.1%) | |
| IIB | 15 (15.3%) | 11 (11.2%) | 4 (4.1%) | |
| IIIA | 10 (10.2%) | 4 (4.1%) | 6 (6.1%) | |
| IIIB | 2 (2.0%) | 1 (1.0%) | 1 (1.0%) | |
| IVa | 2 (2.0%) | 2 (2.0%) | 0 (0.0%) | |

Postoperative data are reported in Table 2. The mean POD at discharge was 6.7 ± 2.6 days, with a median postoperative hospital stay of 6 days (from 4 to 20 days). The SDC group showed a significantly lower postoperative hospitalization (mean 6.1 ± 2.0 days vs. 7.3 ± 3.1; median 6.0 ± 2.0 vs. 6.5 ± 2.8 days; *p* = 0.02). Overall, this was significantly longer in patients who presented at least one postoperative complication (*p* = 0.01). The most frequent postoperative complication was persistent air leakage (PALs, 9.2%), followed by pneumonia (4.1%) and atrial fibrillation (2%). There was no difference between ST and SDC groups in incidence of postoperative complications (*p* = 0.67). Thirty-six patients (36, 7%) were scheduled for preventive ICU stay after surgery (22 ST and 14 SDC; *p* = 0.13). The mean ICU stay was 1.17 ± 0.51 days, without any difference between the SDC and ST groups (*p* = 0.30). The mean chest tube stay was 5.0 ± 2.0 days (median 5.0 ± 1.0) with SDC group showing a shorter chest drain use (mean 4.7 ± 1.9 days vs. 5.3 ± 2.2 days; median 4.0 ± 2.0 days vs. 5.0 ± 2.0; *p* =0.04).

During the first POD, air leaks were observed in 15 (30%) and 12 (25%) patients in ST and SDC group (*p* = 0.58) respectively. Air leak decreased down to 40% and 58% respectively after three days. PALs, defined as persistence of air leaks for more than 5 days, were recorded in 9 (9.2%) patients without any difference between the two groups (5 ST, 4 SDC; *p* = 0.76). The mean hospital stay in the presence of PALs was 10.4 ± 3.4 days (median 9.0 ± 1.0) versus 6.3 ± 2.3 (median 6.0 ± 1.0) days in patients without PALs (*p* = 0.007). PALs were treated in two (22.2%) patients with blood patch and conservatively in the remaining.

The SDC group showed a lower total fluid drainage (median 1150.0 ± 651.5 mL vs. 1477.5 ± 762.5 mL; *p*-value = 0.07). This difference was statistically significant in POD 1 (median 400.0 ± 200.0 mL vs. 450.0 ± 193.8 mL, *p* = 0.04) and as a mean of first three PODs (median 325.0 ± 137.5 mL vs. 362.5 ± 96.7 mL, *p* = 0.01). Fluid drainage was also assessed at CXR using a 4-grade scale; no difference was observed between groups on POD 1 (ST 1.3 vs. SDC 1.1; *p* = 0.34), POD3 (ST 1.0 vs. SDC 0.9; *p* = 0.76) and after tube removal (ST 1.3 vs. SDC 1.2; *p* = 0.65). The pleural fluid retention rate, defined as percentage of patients with grade 2 or 3 at CXR evaluation, showed no difference between the two groups in POD 1 (ST 32% vs. SDC 18.8%; *p* = 0.13), POD 3 (ST 26% vs. SDC 27%; *p* = 0.90) and after tube removal (ST 40% vs. SDC 37.5 %; *p* = 0.80).

**Table 2.** Post-operative characteristics. Data are number (%) or mean ± standard deviation. ICU = Intensive Care Unit; POD = postoperative day; *p*-value ≤ 0.05 are considered statistically significant.

| | Overall | Standard | Coaxial | *p*-Value |
|---|---|---|---|---|
| Length of stay (days) | 6.7 ± 2.6 | 7.3 ± 3.1 | 6.1 ± 2.0 | 0.02 |
| Tube stay (days) | 5.0 ± 2.0 | 5.3 ± 2.2 | 4.7 ± 1.9 | 0.04 |
| **Postoperative complications** | | | | |
| Overall | 18 (18.4%) | 10 (10.2%) | 8 (8.2%) | 0.67 |
| Persistent air leaks | 9 (9.2%) | 5 (5.1%) | 4 (4.0%) | 0.76 |
| Sputum retention | 4 (4.0%) | 2 (2.0%) | 2 (2.0%) | |
| Atrial Fibrillation | 2 (2.0%) | 2 (2.0%) | 0 (0.0%) | |
| Others | 3 (3.1%) | 1 (1.0%) | 2 (2.0%) | |
| ICU admission | 36 (100%) (36.7%) | 22 (22.4%) | 14 (14.3%) | 0.13 |
| ICU stay (days) | 1.2 ± 0.7 | 1.2 ± 0.6 | 1.1 ± 0.3 | 0.30 |
| **Air leaks detection** | | | | |
| POD 1 | 27 (27.5%) | 15 (15.3%) | 12 (12.2%) | 0.58 |
| POD 3 | 14 (14.3%) | 9 (9.2%) | 5 (5.1%) | |
| **Amount of drainage (mL)** | | | | |
| Overall | | 1624.9 ± 718.5 | 1363.5 ± 692.2 | 0.07 |
| POD 1 | | 464.4 ± 143.0 | 407.9 ± 141.4 | 0.04 |
| POD ≤ 3 | | 374.2 ± 96.1 | 323.9 ± 94.5 | 0.01 |
| **Chest X-ray scale (grade)** | | | | |
| POD1 | | 1.3 ± 0.8 | 1.1 ± 0.8 | 0.34 |
| POD3 | | 1.0 ± 0.9 | 0.9 ± 1.0 | 0.76 |
| Post-removal | | 1.3 ± 0.8 | 1.2 ± 1.1 | 0.65 |
| **Fluid retention rate (scale)** | | | | |
| POD1 | | 16 (16.3%) | 9 (9.2%) | 0.13 |
| POD3 | | 13 (13.3%) | 13 (13.3%) | 0.90 |
| Post-removal | | 20 (20.4%) | 18 (18.4%) | 0.80 |
| **Pain (Visual Analogue Scale)** | | | | |
| POD 1 | | 5.5 ± 1.9 | 4.6 ± 1.7 | 0.02 |
| POD 3 | | 4,0 ± 1.5 | 4.2 ± 1.8 | 0.70 |
| POD 5 | | 2.8 ± 1.6 | 2.4 ± 1.2 | 0.14 |
| **Tube Removal Complications** | | | | |
| Overall | 22 (22.4%) | 13 (13.3%) | 9 (9.2%) | 0.47 |
| Pneumothorax | 14 (14.3%) | 8 (8.2%) | 6 (6.1%) | |
| Pleural Effusion | 5 (5.1%) | 3 (3.1%) | 2 (2.0%) | |
| Hydro-pneumothorax | 2 (2.0%) | 1 (1.0%) | 1 (1.0%) | |
| Subcutaneous emphysema | 1 (1.0%) | 1 (1.0%) | 0 (0.0%) | |

The mean postoperative pain, measured with the VAS on the first, third and fifth POD, was 4.1 ± 1.3 for the ST group and 3.7 ± 1.3 for the SDC group, without significant difference (*p* = 0.11). However, the ST group significantly complained of higher pain in POD1 (5.5 ± 1.9 vs. 4.6 ± 1.7; *p* = 0.02). Thirty-nine patients (39.8%) required the administration of supplementary analgesic drugs and 25 of them (64.1%) were of ST group (*p* = 0.03).

Complications after chest tube removal occurred in 22 patients (22.4%) (ST 13 vs. SDC 9) without statistical significance between the two groups (*p* = 0.47). The most frequent complication was pneumothorax (14.3%) followed by pleural effusion (5.1%), hydropneumothorax (2%) and subcutaneous emphysema (1%). No one of these occurrences required additional procedures. No re-admissions were recorded in the 30 days after discharge.

Moreover, we have investigated the economic impact including the costs of each tube, hospitalization and the amount of supplemental drugs administration for pain control. As

reported in Table 3, we have observed a significant difference between the two groups with the SDC group showing lower costs ($p = 0.04$).

**Table 3.** Cost analysis. Costs are indicated in euro. Total cost = (hospital daily cost + cost of drugs) × mean length of stay + chest tubes cost (single device).

|  | Standard | Coaxial | Mean Difference | *p*-Value |
|---|---|---|---|---|
| Chest tubes cost | 21.7 | 64.5 | 42.8 | |
| Drugs cost (mean) | 16 | 15.9 | 0.1 | |
| Hospital cost per days [8] | 674 | 674 | 0 | |
| Mean length of stay (days) | 7.3 | 6.1 | 1.2 | |
| Total cost | 5059 | 4273 | 786 | 0.04 |

## 4. Discussion

Historically, textbooks recommended the use of two chest tubes after major pulmonary resections: one placed inferiorly to drain fluids and one towards the apex to facilitate lung expansion [9,10].

Recently, many studies reported that a single chest tube could be adequate [11]. Randomized clinical trials [1–4] compared the use of one chest tube with the standard two chest tubes in patients undergoing lobectomy disclosing no statistically significant differences in hospital stay, pleural drain capability and post-removal complications.

Moreover, recent meta-analysis [12–14] showed that the use of a single chest drain is more effective than two to reduce postoperative pain and to facilitate patients' compliance to postoperative physiotherapy. This leads to a shorter hospitalization and reduction of costs. Despite that, some institutions still prefer to insert two tubes after thoracotomy to optimize fluid and air drainage [15,16]. In fact, there is still reserve in using a single chest tube after open thoracic procedure because of the possibility of avoiding suboptimal pleural space management. Moreover, there are still concerns about tube clotting, which is observed in up to 5.8% of the patients with one chest tube [5,17], and other complications such as loculated pleural effusion and inefficient fluid drainage of the costophrenic angle. At the same time, none of the clinical studies have shown the superiority of two standard chest tubes over a single chest tube after pulmonary lobectomy. In this regard, we have assumed that double chest tubes are still the most representative standard of practice worldwide.

Recently, a new flexible coaxial drain was developed to combine the benefits of two separate chest drains with the proven advantages of one chest tube. It is made of biocompatible silicone and is composed by four external fluted channels for fluids drainage and an internal section which allows separate air evacuation from appropriate distal bores. Compared with STs, the draining surface area provided by SDC is considerably wider and resistant to clot occlusion. Furthermore, Guerrera et al. showed that SDC provides a satisfactory air evacuation even in patients with significant air leaks [18].

In 2017, Rena et al. [5] retrospectively compared 52 patients treated with SDC with 104 patients with the standard two chest tubes after open or VATS lobectomy: SDC resulted "non-inferior" in fluid and air evacuation, hospital stay and rate of postoperative complications. However, one of the limitations of that study was the retrospective nature and the absence of randomization.

This is the first randomized clinical trial comparing the use of two standard tubes with one SDC. SDC resulted as an effective option after pulmonary lobectomy. Regarding fluid drainage, the SDC group showed a lower fluid evacuation compared to ST group, in particular during the first three PODs. However, at CXR there was no difference between the two groups in terms of pleural effusion, suggesting that one SDC tube provides sufficient cleaning of the chest cavity. Indeed, the draining surface provided by SDC is wider than both superior and inferior STs. Thus, the higher fluid evacuation provided by two STs, estimated at approximately 50 mL per day, might be the effect of the pleural reaction to the presence of a double big bore catheter.

Concerning air aspiration, SDC appears "non-inferior" to STs. The air leaks rate is similar between the groups, as well as the rate of PALs. SDC provides adequate air evacuation even in the presence of high flux air leaks. The rate of fixed pleural space [19], defined as incomplete re-expansion of the lung after resection in absence of air leaks, is similar in the two groups and it seems more related to patients' characteristics than to inadequate air evacuation.

In our series, SDC group showed a significantly shorter hospitalization. This could be explained with the tendency of patients to promptly adhere to mobilization and physiotherapy. In fact, even if the mean postoperative pain showed no significant differences between the groups, the SDC group showed a significantly lower pain in POD1. This topic is crucial in the era of enhanced recovery after surgery (ERAS). Late mobilization has been proven to be an independent factor correlated to delayed discharge and increased morbidity [20]. Furthermore, ST patients more often required additional analgesic drugs: this implementation suggests inadequate pain control despite the fact that we didn't find a difference in VAS score. However, the comparison between a double chest tube versus a single chest tube technique might be quite obviously associated to a better outcome after a single chest tube. This assessment could lead to a future research interest for comparing single SDC versus a single standard chest tube.

Finally, although costs of single SDC are remarkably higher compared to STs, the shorter hospital stay and lower analgesic drugs administration drop the total costs.

This study presents some limitations: although it is a randomized study, the number of patients is relatively small. Different surgeons performed the surgical procedures with no standardization in the use of hemostatic agents. We do not compare the SDC tube with a single standard tube. The study is focused on open lobectomies and the supposed advantages of SDC tubes in VATS procedures need confirmation in other studies. Finally, this is a single center experience and larger multicenter studies are required to confirm our results.

## 5. Conclusions

Our results suggest that SDC might be a feasible option after open pulmonary lobectomy with similar results in terms of fluid draining capability, air suction and post removal complications compared to two standard chest tubes. Furthermore, the SDC group showed a lower analgesic drugs requirement, lower postoperative pain in POD1 and a shorter hospital stay and thus a reduction of costs.

**Author Contributions:** M.B.: Conception and design, Data analysis and interpretation, manuscript writing, final approval of manuscript; E.M.: Data analysis and interpretation, manuscript writing, final approval of manuscript; S.M.: Provision of study materials, Collection and assembly of data, final approval of manuscript; D.A.: Administrative support, Data analysis and interpretation, final approval of manuscript; A.P.: Administrative support, Data analysis and interpretation, final approval of manuscript; D.D.: Administrative support, Data analysis and interpretation, final approval of manuscript; C.P.: Provision of study materials, Collection and assembly of data, final approval of manuscript; J.V.: Provision of study materials, Collection and assembly of data, final approval of manuscript; T.D.G.: Conception and design, Data analysis and interpretation, final approval of manuscript; E.A.R.: Conception and design, Data analysis and interpretation, manuscript writing, final approval of manuscript; F.V.: Conception and design, Data analysis and interpretation, manuscript writing, final approval of manuscript; M.A.: Conception and design, Data analysis and interpretation, Collection and assembly of data, manuscript writing, final approval of manuscript. All authors have read and agreed to the published version of the manuscript.

**Funding:** This research received no external funding.

**Institutional Review Board Statement:** The study has been approved by the Ethics Committee of Policlinico Umberto I, Department PARIDE STEFANINI on 26 January 2017 (ref. number 1/2017) and and then refreshed in 2020 to update the spectrum of data ref number 02/2020.

**Informed Consent Statement:** Informed consent was obtained from all subjects involved in the study.

**Data Availability Statement:** The data presented in this study are available on request from the corresponding author.

**Conflicts of Interest:** The authors declare no conflict of interest.

**Abbreviations**

| | |
|---|---|
| SDC | Smart Drain Coaxial |
| ST | Standard tubes |
| ICU | Intensive care unit |
| VAS | Visual analogue scale |
| CXR | Chest X-ray |
| POD | Postoperative day |
| IQR | Inter-quartile range |
| SD | Standard deviation |
| PALs | Prolonged air leaks |
| ERAS | Enhanced recovery after surgery |

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
