# Peer review of "Coaxial Drainage versus Standard Chest Tube after Pulmonary Lobectomy: A Randomized Controlled Study"

_curroncol, doi:10.3390/curroncol29070354_

Round 1

Reviewer 1 Report

I would like to thank the Authors for their work which aims at comparing in a randomised setting the coaxial drainage vs. a standard chest tube following a lobectomy.

The topic is of relative interest to the thoracic surgeons given the amount of literature available on chest drains in the post-lobectomy setting and on the coaxial drain. 

Furthermore, this kind of drain may not be readily available and, even in presence of data, it usually is still is up to the surgeons preference whether or not use one or more drains and the personal inclination towards a change in the type of drain to be used. 

However, this study draws a line in favour of the coaxial drain on data produced by a RCT in a debatable setting.

In fact, it could be a flaw to assume that two chest drains are the standard for lobectomy after thoracotomy.

The Authors have given questionable reasons for that assumption in the discussion “Four randomised clinical trials (1 – 4) compared the use of one chest tube with the standard two chest tubes in patients undergoing lobectomy and/or bilobectomy. All the studies concluded that one chest tube was “non-inferior” compared to two chest  tubes, without statistically significant differences in terms of hospital stay, pleural drain capability and post-removal complications”. 

Based on this affirmation, the decision of using two drains is controversial, especially in the setting of a RCT, and may give room to speculating that the use of two drains after an open lobectomy is actually a a standard procedure of theirs', onto which was structured the RCT, rather than theorising on the outdated data (2016-2017) to which the Authors refer to. 

For such reasons, it would have been more correct and appealing to compare the coaxial drain to a single standard chest drain, or eventually, two coaxial drains with two standard drains.

Also, that would have allowed the inclusion of VATS procedures, which are in this paper an exclusion criteria and therefore a loss of a probably more valuable piece of information, given the fact that VATS approaches are being adopted worldwide as a standard approach, at least in the early stages.

This must be acknowledged in the paper, possibly a "limitations and strengths" paragraph. 

However, the paper is generally well written, the M&M and results are well described and is a properly structured RCT. 

It would be interesting if the Authors could add to the paper the answers to the following questions, which aim at defining the clinical standardisation, which could possibly interfere with the results:

  • Authors state that “In all the patients, the regular lobectomy plus standard lymphectomy was performed without intraoperative complications and showed no significant anatomical variances”. However, it would be interesting to know the cTNM and Stage of each procedure in order to assess if a more radical lymphadenectomy was necessary based on cN (example lobectomy + lymphadenectomy of single station N2). Please acknowledge.
  • Did the surgeons use any sealants/glues/fibrin sponge (others)  during the procedure? If so, please state them.
  • Is their use standardised or is it based onto the surgeons’ preference during the procedure? Please acknowledge.
  • Were digital drains or standard water seal drains used? Please acknowledge.
  • Was any energy device utilised routinely? Please acknowledge.
  • Complications occurred in both groups without being statistically significant and with no need of additional procedures: was any patient readmitted post-discharge due to any complication which could be related to the chest drains? Please acknowledge.

The paper could be eligible for publishing following a revision comprehensive of the requested acknowledgements.

Author Response

Thank you very much for your supportive comments and suggestions. 

Please deal with the following answers aimed at improving the quality of submission.

Comment 1: Authors state that “In all the patients, the regular lobectomy plus standard lymphectomy was performed without intraoperative complications and showed no significant anatomical variances”. However, it would be interesting to know the cTNM and Stage of each procedure in order to assess if a more radical lymphadenectomy was necessary based on cN (example lobectomy + lymphadenectomy of single station N2). Please acknowledge.

Answer 1: We see the point and we agree with the reviewer. The Lung cancer staging, according to the 8th edition of the IASLC (International Association for the Study of Lung Cancer) lung cancer staging system, was added as suggested in Result section and in Table 1.

Comment 2: Did the surgeons use any sealants/glues/fibrin sponge (others)  during the procedure? If so, please state them. Is their use standardised or is it based onto the surgeons’ preference during the procedure? Please acknowledge.

Answer 2: In our institution we generally use fibrin sealant on the vascular and bronchial stumps but the use of other hemostatic agents like fibrin sponges or oxidised regenerate cellulose is up to surgeons preference and intraoperative issues and there is not a standardized technique. A sentence was added in the Results section to state that.

Comment 3: Were digital drains or standard water seal drains used? Please acknowledge.

Answer 3: We routinely use standard water seal. This information was added as suggested in the M&M section.

Comment 4: Was any energy device utilised routinely? Please acknowledge.

Answer 4: In our institute we use routinely energy devices only in VATS procedures while in open surgery it is generally not necessary.

Comment 5: Complications occurred in both groups without being statistically significant and with no need of additional procedures: was any patient readmitted post-discharge due to any complication which could be related to the chest drains? Please acknowledge.

Answer 5: We had no re-admission after discharge. This information was added as requested in the Results section.

Reviewer 2 Report

Thank you for submitting this article. I was pleased to receive it as a reviewer.

I have some queries and suggestions that you should address first.

First, the paper should be rewritten according to the Consolidated Standards of Reporting Trials (CONSORT) statement [www.consort-statement.org/]. In particular, a CONSORT checklist should be added [http://www.consort-statement.org/consort-statement/checklist]; a template of the CONSORT flow diagram is available in MS Word [http://www.consort-statement.org/download/Media/Default/Downloads/CONSORT%202010%20Checklist.doc].

Secondly, the statistical analysis should be written according to the recently published guidelines [Hickey GL, Dunning J, Seifert B, Sodeck G, Carr MJ, Beyersdorf F on behalf of the EJCTS and ICVTS Editorial Committees Editor's Choice: Statistical and data reporting guidelines for the European Journal of Cardio-Thoracic Surgery and the Interactive CardioVascular and Thoracic Surgery. Eur J Cardiothorac Surg 2015;48:180-93].

Please also comment about the sample size determination [Hickey GL, Grant SW, Dunning J, Siepe M. Statistical primer: sample size and power calculations—why, when and how? Eur J Cardiothorac Surg. 2018;54(1):4-9.].

Please better describe the limitations of the study in the discussion.

The discussion should be improved with a better search of the literature.

About minor points, there are typo and errors in the text. Please thoroughly check the article.

Good luck with your article, and thanks again for submitting it.

Author Response

Thank you for your comments aimed to improve the quality of manuscript. Please find below the changes made following your useful suggestions.

Comment 1: First, the paper should be rewritten according to the Consolidated Standards of Reporting Trials (CONSORT) statement [www.consort-statement.org/]. In particular, a CONSORT checklist should be added [http://www.consort-statement.org/consort-statement/checklist]; a template of the CONSORT flow diagram is available in MS Word [http://www.consort-statement.org/download/Media/Default/Downloads/CONSORT%202010%20Checklist.doc].

Answer 1: The paper was revised according to CONSORT statement. CONSORT checklist has been uploaded as supplemental material.

Comment 2: Secondly, the statistical analysis should be written according to the recently published guidelines [Hickey GL, Dunning J, Seifert B, Sodeck G, Carr MJ, Beyersdorf F on behalf of the EJCTS and ICVTS Editorial Committees Editor's Choice: Statistical and data reporting guidelines for the European Journal of Cardio-Thoracic Surgery and the Interactive CardioVascular and Thoracic Surgery. Eur J Cardiothorac Surg 2015;48:180-93].

Answer 2: The statistical analysis section was improved and changed according to the recently published guidelines.

Comment 3:Please also comment about the sample size determination [Hickey GL, Grant SW, Dunning J, Siepe M. Statistical primer: sample size and power calculations—why, when and how? Eur J Cardiothorac Surg. 2018;54(1):4-9.].

Comment 3: Sample size calculation data have been added in the statistical analysis section as requested.

Comment 4: Please better describe the limitations of the study in the discussion.

Answer 4: The body of the text has been changed improving the description of the limitations of the study.

Comment 5: The discussion should be improved with a better search of the literature.

Answer 5: Discussion section has been improved with other relevant research on this topic.

Comment 6: About minor points, there are typo and errors in the text. Please thoroughly check the article.

Answer 6: The text has been checked and errors corrected.

Round 2

Reviewer 1 Report

I would like to thank the Authors for the requested changes made to the manuscript.

Yet, I would suggest the Authors to provide further modifications to the introduction and the first part of the discussion.

Too much focus is given onto how many drains should be placed in a chest after lobectomy and what literature says on that topic.

To the reader this is confusing and could sound like an unnecessary justification. 

In concept, in order to facilitate the reader to focus onto the proven advantages of this coaxial drain, Authors could simply state that traditionally they use one apical drain (air) and a basal one (fluid) for open lobectomies and in order to shift to a single drain they tested a drain with specifications which, apparently, does both (SCD). 

For that reason, lines 36-39 of the introduction and lines 186-197 of the discussion, are irrelevant and confusing.

Furthermore, the manuscript needs English editing (e.g. "The mean chest tube maintenance was..." page 4, Line 143).

With such improvements, the manuscripts will be ready for publishing.

Author Response

Thank you for your further suggestions.

The introduction part has been erased for a quicker reading as suggested. As concern the discussion section, we took into account your comment along with the other reviewer's suggestions, which asked to expand the discussion with more literature on this topic. We have tried our best to fulfill both requests. We hope you will find satisfactory our changes. 

English has been reviewed and corrected.

Reviewer 2 Report

No further comments

Author Response

We would like to thank you again you for your suggestions. 

Round 3

Reviewer 1 Report

I would like the Authors for their effort in updating the manuscript as suggested by the reviewers. 

I believe the paper is eligible for publishing.

Best.

Reviewer 2 Report

No further comments